# BIreactive: Expanding the Scope of Reactivity Predictions to Propynamides

**DOI:** 10.3390/ph16010116

**Published:** 2023-01-12

**Authors:** Markus R. Hermann, Christofer S. Tautermann, Peter Sieger, Marc A. Grundl, Alexander Weber

**Affiliations:** Medical Chemistry, Boehringer Ingelheim Pharma GmbH & Co. KG, Birkendorferstr. 65, 88397 Biberach an der Riß, Germany

**Keywords:** targeted covalent inhibitors, drug discovery, covalent warheads, reactivity assessment, glutathione, propynamide

## Abstract

We present the first comprehensive study on the prediction of reactivity for propynamides. Covalent inhibitors like propynamides often show improved potency, selectivity, and unique pharmacologic properties compared to their non-covalent counterparts. In order to achieve this, it is essential to tune the reactivity of the warhead. This study shows how three different in silico methods can predict the in vitro properties of propynamides, a covalent warhead class integrated into approved drugs on the market. Whereas the electrophilicity index is only applicable to individual subclasses of substitutions, adduct formation and transition state energies have a good predictability for the in vitro reactivity with glutathione (GSH). In summary, the reported methods are well suited to estimate the reactivity of propynamides. With this knowledge, the fine tuning of the reactivity is possible which leads to a speed up of the design process of covalent drugs.

## 1. Introduction

Covalent drugs have been known as therapeutic agents for decades and are successfully applied for several diseases with different indications [1,2,3]. Identification of covalent drugs in the last century had a serendipity-driven component, and the mode of action was often confirmed after release to the market and patient treatment [4]. Aspirin (Figure 1, compound **I**), penicillin (**II**), and omeprazole (**III**) are classical and highly successful examples of covalent drugs discovered serendipitously. Despite these early successes, further rational exploration of covalent drugs in the past stagnated, arguably due to findings associating highly reactive metabolites with toxicity findings, raising concerns about the use of reactive functional groups in covalently binding drugs. However, recent analysis indicates no major disadvantages of approved covalent drugs with respect to severe side effects compared to non-covalent drugs [5]. Successful market entries and recent developments, including the successful targeting of “undruggable” targets such as KRAS, led to a resurgence of covalent drugs. General benefits include high potency due to the high binding energy contribution of the covalent bond and (for irreversible covalent binders) an extended pharmacodynamic effect, decoupled from the pharmacokinetic properties of the compound [2,6].

The analysis and inclusion of target structure information support the identification of new covalent drugs, including structure-based design and virtual screening approaches [7,8,9,10]. In addition, successful screening against KRAS G12C with further optimization cycles led to a clinical candidate, including structure-based design efforts in combination with targeted compound library synthesis [11,12]. From a mechanistic point of view, covalent inhibitors interact with a target protein in a two-step process (Figure 1). In a first step, covalent inhibitors interact with the target via non-covalent interactions like hydrogen bonds, dispersion and hydrophobic interactions, and salt bridges. This brings the reactive functional group, the so-called “warhead”, in close proximity to a reactive function within the target protein, e.g., the thiol function of a cysteine. Hence, in a second step, reactions between these reactive functions lead to the formation of a covalent bond between the ligand and the protein target [13,14,15,16,17,18,19]. The so-called targeted covalent inhibitors (TCI) were designed to address catalytic or non-catalytic nucleophiles with their electrophilic warheads through either a reversible or irreversible mechanism [20,21,22,23,24,25]. To reduce the potential for toxicity, often associated with unspecific covalent binding, a balanced warhead reactivity is a primary design goal for TCIs [18]. Currently, several TCIs are on the market with valuable target product profiles [5]. The clinical success of TCIs was initiated with the covalent inhibition of enzymes of the protein kinase family [26]. A selected subset of covalent drugs on the market or in clinical development is shown in Figure 1, targeting the epidermal growth factor receptor (EGFR) [21], Bruton’s tyrosine kinase (BTK) [6,22,23,24], Janus kinase 3 (JAK3), and other TEC kinase family members with a cysteine residue in the binding pocket [25]. Osimertinib (**IV**) is targeting EGFR, whereas ritlecitinib (**VI**) is covalently addressing JAK3 and other TEC kinase family members. The remaining TCIs (**V**, **VII**, **VIII**, and **IX**) are covalent inhibitors of BTK. Whereas osimertinib (**IV**), ibrutinib (**V**), and ritlecitinib (**VI**) contain an acrylamide warhead to address a non-catalytic cysteine residue, next-generation BTK inhibitors include a propynamide warhead functional group. Acalabrutinib (**VIII**) and tirabrutinib (**VII**) were launched in 2017 and 2020, whereas branebrutinib (**IX**) is currently in Phase II clinical trials, showing the value of the propynamide warhead for covalent drug design.

To tailor and estimate warhead reactivity of covalent drugs, experimental setups are used, however with the limitations of compound availability and limited throughput of experimental testing [27]. For the prospective design of covalent inhibitors, computational approaches were introduced to speed up and fine-tune warhead reactivity by making reliable predictions [28,29,30]. The correlation with experimentally derived reactivity data is used to judge the predictiveness of calculated warhead reactivities. As such, assays based on reactivity in the presence of glutathione (GSH) are often applied due to the physiological relevance of the endogenous nucleophile GSH to clear electrophiles, e.g., reactive electrophilic metabolites, via the formation of highly polar GSH adducts. Mimicking physiological conditions, the half-life of adduct formation in the presence of excess GSH under pseudo-first-order kinetic conditions is often used as a measure of the reactivity for warheads within covalently bound compounds, which can be used to estimate the predictiveness of computational approaches [28,29,31,32]. Whereas predictions correlate well with experiment for certain compound subsets, for example the use of Hammett parameters for aromatic acrylamides [33] or the electrophilicity index [34] for unsubstituted acrylamides [35,36], those approaches are limited to certain warhead substitution patterns or warhead classes [37]. Of more general use are quantum mechanical (QM)-derived adduct formation energies or QM transition state energy calculations. Such QM-derived reactivity predictions were successfully applied for different warhead classes [38,39]. In addition, large numbers of computationally demanding QM calculations could be the basis for machine-learning models to further increase the number of compounds for reactivity prediction with shorter cycle times [37]. In this work, we describe the expansion of our previously reported “BIreactive” approach for propynamides, a covalent warhead class included in clinical candidates and approved drugs on the market.

**Figure 1 pharmaceuticals-16-00116-f001:**
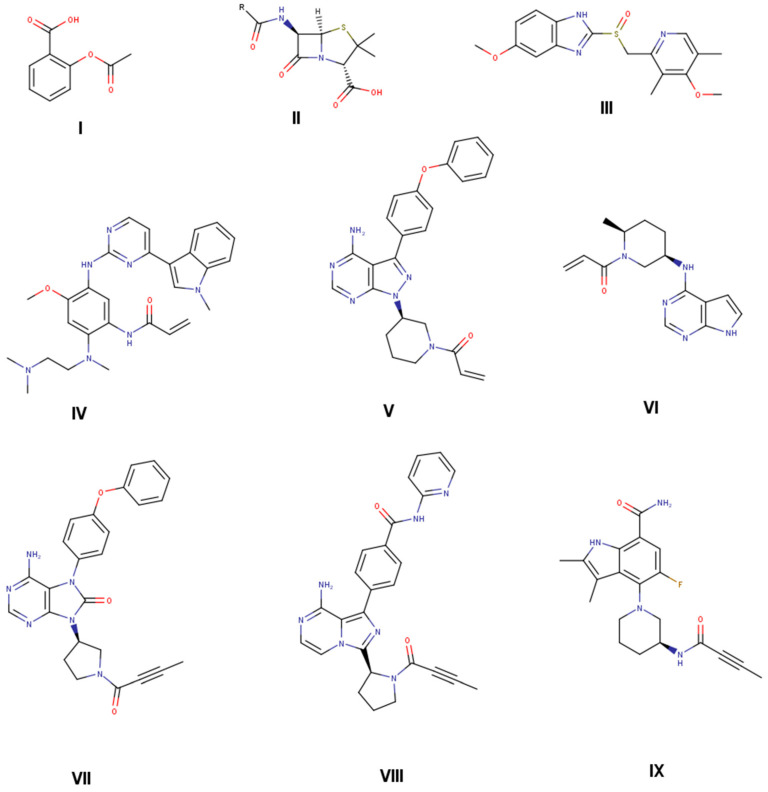
Selected approved drugs that are on the market with covalent mode of action identified after product launch (aspirin (**I**), penicillin G (**II**), and omeprazole (**III**) [18,37]). Selected targeted covalent inhibitors (TCIs) on the market designed to covalently interact with target protein (osimertinib (**IV**) [21], ibrutinib (**V**) [22], ritlecitinib (**VI**) [25], tirabrutinib (**VII**) [23], acalabrutinib (**VIII**) [24]), and in phase II clinical trials (branebrutinib (**IX**) [6]).

## 2. Materials and Methods

Calculations: The calculations were done similarly to our recent paper on acryl amides [37]. To mimic the reaction of the warhead, GSH methanethiolate (CH_3_S^−^) was used as a reactant. All calculations were performed with Gaussian16 [40] using ωB97XD/cc-pVDZ [41,42,43] as the level of theory, and CPCM to take solvent effects into account [44,45]. A truncation algorithm according to reference [35] for the molecular structures was used to speed up the calculations. In the same publication, it was shown that the truncation algorithm has no effect on predictability. To generate a reasonable ensemble of conformers, we generated one conformation with CORINA [46,47] and added four additional conformations using OMEGA [48,49]. To test the reliability of the small double zeta basis set, single-point calculations with cc-pVQZ on the structures gained by ωB97XD/cc-pVDZ were performed (Appendix A).

In the first step, for each compound, the structure of each conformation is optimized by DFT. From this structure, the electrophilicity index ω  is calculated.
$$\omega =\frac{\begin{array}{c} {\left(\frac{1}{2}(E_{\mathrm{LUMO}}+E_{\mathrm{HOMO}}\right)}^{2} \end{array}}{E_{\mathrm{LUMO}}-E_{\mathrm{HOMO}}}$$
for the calculation of the electrophilicity index, the LUMO and HOMO orbitals with high localization on the warhead are chosen. Both pi-orbital combinations (in the N—C—O plane and perpendicular) are investigated. Only the perpendicular results are shown. The in-plane orbitals show a slightly worse correlation.

To calculate the transition state structure, a methanethiolate is placed at 2.4 Angstroms from the electrophilic carbon atom and at a 90° angle from the propyl carbon atoms of the optimized structure. All but the warhead atoms and the methanethiolate atoms are positionally constrained. This structure is minimized using MMFF99s [50] within RDKit [51]. This first guess of the TS structure is used to generate additional potential TS structures with different angles for sulfur to attack the electrophilic carbon. There is the possibility for an attack from the side (TS1, O—C—C—S-angle: 90°), from the bottom (TS2, O—C—C—S-angle: 180°), and from the top (TS3, when the oxygen defines the top, O—C—C—S-angle: 0°) (Appendix A). Additionally, the carbon atom from the methanethiolate can point away from the carbonyl (TS_a, C(=O)—C—S—C-angle: ~180°), towards the carbonyl (TS_c, C(=O)—C—S—C-angle: ~0°), or in between these extremes (TS_b, C(=O)—C—S—C-angle: ~90°). Subsequently, a restrained optimization (C—S bond fixed at 2.4 Angstrom) of the described complexes is performed by the MMFF99s force field, followed by a restrained DFT optimization (level of theory as described above). After the preoptimization, a transition state search is initiated to identify the energetically most favored transition state (Figure 2). For exemplary compounds, the intrinsic reaction coordinate was followed to confirm the correct product. For all compounds, the eigenvector corresponding to the imaginary frequency in the transition state agrees with the reaction mechanism, showing a displacement towards the electrophile.

To calculate the adduct formation energy, methanethiolate is added to the propylamine, and sulfur is connected to the electrophilic carbon of the propargyl. The resulting molecule is preoptimized with MMFF99s and RDKit. Subsequently, energy minimization by DFT is performed.

All structures are verified as minima or transition states on the potential energy surface by calculating the Hessian. Thermodynamic contributions are calculated at 298.15 K and 1 atm with the ideal gas phase approximation. The final energies contain zero-point vibrational energy, translational, and rotational contributions to enthalpy and free enthalpy, and will be referred to throughout the whole manuscript.

The ensemble energy over all conformations of a compound is calculated as the Boltzmann average over all conformations for educts, products, or transition states.

In total, approximately 118,000 quantum mechanical calculations were carried out, resulting in GSH activity predictions for 1219 unique molecules.

To establish methods that predict the reactivity of propynamides, a small benchmark data set of 28 molecules was generated, with four different variants of the warhead and seven different scaffold molecules (Figure 3). Out of these 28 molecules, 12 are synthesized, and their half-lives with GSH are measured. This benchmark dataset was expanded with more drugs like propynamides extracted from PubChem [52,53]. More structural diversity was added by generating virtual compounds, starting from the PubChem dataset. For this, an internal matched molecular pair algorithm [54] was used to implement transformation rules [55] to generate variations of the non-covalent part of the PubChem dataset. For PubChem molecules and benchmark molecules, results are available in the Appendix A.

Glutathione reactivity assay: GSH adduct formation data were determined by an inline kinetic HPLC/UV/MS-based assay. A 10 mM DMSO compound stock solution was diluted by a factor of 1:20 with (a) 50 mM phosphate buffer with pH 7.4 and (b) 50 mM phosphate buffer with pH 7.4 plus a 10-fold molar excess of GSH and incubated at 37 °C for different time periods. For (a) 0, 120, and 360 min (providing stability data in plain buffer medium) and for (b) 0, 2.5, 5, 10, 60, 120, 240, and 360 min (providing stability data in the presence of a 10-fold molar excess of GSH).

The decay of the compound is followed by HPLC/UV, and MS is used for parent and GSH adduct identification. Assuming first-order kinetics, half-lives are calculated using Arrhenius plots in both media, where k is obtained from the slope of the graph of the logarithm of the peak area (UV-signal) as a function of time.

## 3. Results and Discussion

In the following section, different transition states are compared for a meaningful discussion, followed by the introduction of a small benchmark dataset where different methods are applied to predict the reactivity of propynamides. In a subsequent analysis, additional and partly public compounds are calculated.

### 3.1. Finding the Transition State

To sample a high variety of different transition states, we started with different input structures to find the transition state with the lowest energy. Several combinations were tested, and it was found that for the investigated ligands, TS1c is the energetically most favorable configuration (Figure 2 and Appendix A; all investigated transition states are depicted in Appendix A). Because of this, in the following, only the results for TS1c are discussed.

### 3.2. Benchmark Dataset

For a first benchmark set, we selected different propynamides that were divided into a right-hand side (RHS), where the warhead carries different substitutions, and a left-hand side (LHS), with different non-covalent scaffolds (Figure 3). The terminal alkyne warhead as in (**1**) is known to have a high degree of reactivity, while the methyl substitution (**2**) leads to less reactive compounds. Additionally, the reactivities for tert-butyl (tBu, **3**) and phenyl (**4**) substituents were analyzed.

To probe the effect of the non-covalent scaffolds, we selected propynamides with aromatic and aliphatic substituents. Additionally, some ring systems and the influence of secondary and tertiary amines are investigated. The propynamides **1**–**4** were synthesized with scaffolds a, b, and e and tested in a GSH assay afterwards (Table 1). All compounds are stable in plain buffer medium.

Angst et al. found a higher reactivity of hydrogen-substituted propynamides when compared to methyl-substituted ones. Their compound with hydrogen has a GSH half-life of 0.1 h compared to 79.6 h for the methyl-substituted variant [56]. We see the same trend here, with hydrogen-substituted propynamides (**1**) being the most reactive compounds, followed by methyl-substituted ones (**2**). Phenyl-substituted warheads show a similar reactivity as their methyl-substituted counterparts (**4**) and the least reactive substitution is the tBu group (**3**).

In a recent publication, propynamides were assessed by the electrophilicity index and showed acceptable correlation with transition state energies [36]. The advantage of the electrophilicity index is that only a single structure optimization of the covalent ligand has to be performed. To follow up this investigation, the electrophilicity index was calculated using the warhead-associated orbitals as described in [33].

The correlation of the electrophilicity index with the experimental GSH values is poor (R^2^: 0.38, Figure 4A). While some correlation may be seen within the methyl- and tBu-warheads, due to the small number of data points, we refrained from a closer inspection. With the electrophilicity index, there are even tBu warheads that are predicted to be more reactive than hydrogen-substituted warheads. This shows that the electrophilicity index is only applicable within one class of compounds [57]. In this study, the aim is to also probe the influence of substitutions on the warhead, and the electrophilicity index is not suitable for this task.

In addition to the electrophilicity index, the transition state energy and adduct formation energy are calculated. The adduct formation energy has the advantage of being calculated with two structure optimizations, for the educts and the product, thus taking roughly double the time of the calculation of the electrophilicity index. Compared to the latter one, the predictability is massively increased (R^2^: 0.81, Figure 4B). Compound 3a shows a too small adduct formation energy in relation to the experimental GSH value. Additionally, the other tBu propynamides are predicted to be a little bit too reactive, but the overall trend is good. The methyl-substituted warheads all follow the same trend, with the terminal alkyne warhead being the most reactive one. One important success factor for the implementation is to focus on the correct conformation of the product. Only by sampling the conformational space of the product can a good correlation be obtained (see Section 2).

The calculation of the transition state energy is the most tedious task here. For such a calculation, the starting structure must be very close to the correct transition state structure; otherwise, the calculation will not converge to the correct transition state. Additionally, the computational demand is higher than for an optimization. Nonetheless, the correlation with the experimental data is very good (R^2^: 0.86, Figure 4C). Hydrogen-substituted warheads have the lowest activation energy E_a_, followed by the methyl- and phenyl-substituted ones. Again, the tBu substituted warheads are predicted to be more reactive than they really are. This trend is illustrated with intrinsic reaction coordinates for compound pairs **1a**/**b**/**e** and **2a**/**b**/**e** in the Appendix A.

One possible reason is that the adduct formation energy and transition state energy overestimate the reactivity of the tBu substituted warheads, which may stem from the truncated methanethiolate (CH_3_S^−^) that was used in this study as a surrogate for cysteine residue of GSH. This may result in an underestimation of steric repulsion, activation energies, and adduct formation energies.

It is striking that for compound **2e**, the transition state energy and the adduct formation energy both predict a much more reactive compound than the experiment.

A direct comparison of the adduct formation energy and the transition state energy shows that both values correlate well with each other (R^2^: 0.76, Figure 4D). As this can be expected for this class, earlier results show that it does not hold true for 2-Chloroacetamides [36].

The correlations between the electrophilicity index and adduct formation energy and transition state energy are shown in the Appendix A. In general, the correlation is very poor.

### 3.3. Results for the Literature Known Compounds

To go beyond a small benchmark dataset, compounds from internal and external sources and virtual compounds with a propynamide group were included in the analysis, as described in the Methods Section. All reported GSH data come from in-house experiments to ensure assay consistency. Results are shown in Figure 5. A CSV file with results can be found in the Appendix A.

The full dataset shows a similar trend as the benchmark dataset. The electrophilicity index has a slightly better correlation with experimental GSH values (Figure 5A) than the benchmark dataset but remains inferior to the other methods discussed herein. Looking at different classes, the R^2^ values are good for phenyl-substituted warheads, mediocre for hydrogen, and bad for methyl substitutions (R^2^ = 0.96, 0.41, and 0.12; R^2^_Overall_ = 0.47).

The adduct formation energy and the transition state energy are again very good at predicting the experimental GSH half-life (R^2^ = 0.82/0.86, Figure 5B,C). For several compounds, GSH reactivity is reported with operator values.

The good correlation between the transition state energy and the adduct formation energy creates confidence that both established models are able to predict the reactivity of propynamides in this dataset (Figure 5D).

### 3.4. Extending the Warhead Substitutions

There is a big influence of the electronegativity of the substituents on the reactivity of the warhead. To further expand the chemical space, seven additional warhead substitutions were chosen that span a wider range of electronic effects on the warhead (Figure 6). The hypothesis is that electron-withdrawing groups enhance the reactivity while electron-donating groups reduce the reactivity.

The calculated results confirm the hypothesis that the reactivity and electronic effect of the substitutions are highly correlated (Figure 7). The most reactive warhead in this subset is the trifluoromethyl substituted propynamide (**5**), which has a strong electron-withdrawing effect, followed by the nitrophenyl substituent (**6**). Substituents with smaller electronic effects like methylphenyl (**9**) and trifluorophenyl (**8**) have a smaller effect, while electron pushing substituents like methoxyphenyl (**10**) and aminophenyl (**11**) reduce the reactivity. This knowledge can be leveraged to specifically fine-tune warhead reactivity.

## 4. Conclusions

Our results demonstrate that the reactivity of propynamides, a warhead class included in several clinical candidates and approved drugs, is of high interest in drug design and can be predicted by in silico methods. A thorough analysis of transition state structures showed that in the most accessible transition state, the sulfur from methanethiolate (CH3S^−^) attacks the warhead perpendicular to the carbonyl–ethynyl plane (Figure 2). Recent publications indicate that the electrophilicity index could be used to predict, for example, the reactivity of acrylamides, another warhead type included in approved drugs [33,35,36]. However, our investigations show limitations when predicting the reactivity of propynamides with different substitutions on the warhead. The reactivity of the propynamide warhead can be best described by the transition state energy and the adduct formation energy.

A comprehensive benchmark for a wide range of substituted propynamides shows a considerable influence of warhead substitutions on reactivities. This effect can be mainly understood by electronic properties. The most reactive warhead features a substitution with a simple hydrogen, followed by methyl- and tert-butyl substituents. Phenyl substitutions show a similar reactivity to methyl substitutions (predicted reactivity: H > CH3~Phenyl > tBu). As a rule of thumb, electron-withdrawing groups increase the reactivity while electron-donating groups reduce the reactivity, and bulky substituents may inhibit the reaction totally.

For substitutions on the scaffold part, aromatic systems are more reactive than aliphatic ones, as illustrated by the benchmark dataset.

We also report 12 experimental values of half-lives in a GSH assay for various propynamides. This data can be used to develop additional methods for the prediction of GSH reactivity. In a follow-up study, we are aiming to use the transition state energies and adduct formation energies to build a machine learning model for the rapid prediction of these properties. This has already been done for acrylamides and can speed up the prediction for propynamides tremendously [37].

Insight into the reactivity of propynamides helps in the design of new TCIs that target, for example, a cysteine nucleophile. The prediction of the reactivity of compounds from the public dataset shows that its applicability domain expands to pharmaceutically relevant molecules. The composed data can be useful for scientists to understand the reactivity of covalent drugs.

## Data Availability

Data is contained within the article and Appendix A.

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
