# Peer review of "BIreactive: Expanding the Scope of Reactivity Predictions to Propynamides"

_pharmaceuticals, 2023, doi:10.3390/ph16010116_

Round 1

Reviewer 1 Report

This paper describes the application of a protocol 5o check the reactivity of propyneamides.

Nevertheless, the scope of the paper was not outlined and disccussed in the main text, with no clear application in drug Discovery process.

All the examples did not reflect experimental data or new drug candidates but focused only on simplex drugs available on the market.

Considering the High pharmacological profile of the journal, I suggest to rewrite the paper and try publication in a more specific computatiomal journal 

Author Response

This paper describes the application of a protocol 5o check the reactivity of propyneamides.

Nevertheless, the scope of the paper was not outlined and disccussed in the main text, with no clear application in drug Discovery process.

Reply: We changed the abstract and hope, the scope is now a little bit clearer.

All the examples did not reflect experimental data or new drug candidates but focused only on simplex drugs available on the market.

Reply: We do not understand the criticism. We published inhouse generated experimental GSH data. Also, it was never our intention to disclose new drug candidates. In the second part we showed the calculation of real drugs on the market. A list of compounds that are calculated and are on the market or in phase 2 clinical trials is shown in Figure 1.

Considering the High pharmacological profile of the journal, I suggest to rewrite the paper and try publication in a more specific computatiomal journal 

Reliable in silico prediction method are very important in the field of covalent drug discovery, and we show the first comprehensive study on propyneamides. With the inclusion of the raw data, scientists can easily compare their methods with ours. This will accelerate covalent drug discoveries. In our opinion, this qualifies the manuscript to be published in Pharmaceuticals. But we leave this decision to the editor.

Reviewer 2 Report

This manuscript reports the computational reactivity prediction of propyneamides against GSH. Considering that propyneamides together with acrylamides represent an important warhead type of targeted covalent inhibitors, developing an in silico tool for their reactivity prediction is highly relevant. Previously the authors developed a useful QM-ML approach ’BIreactive’ for the reactivity prediction of acrylamides and 2-chloroacetamides. This approach has been now extended towards propyneamides. First, the transition state geometry against MeS- was investigated and next they calculated transition state energies and the adduct formation energies that were correlated to experimental GSH half-lives. The results confirmed that the methodology is capable describing the reactivity and therefore it can be useful for designing new covalent ligands with propyneamide warhead. There are some minor issues to be addressed before accepting the paper for publication.

1.           The authors showed that electrophilicity indices are less useful for the prediction of reactivity for this class of compounds. This observation, however, is not new. There is a general consensus in the literature that electrophilicity index can be only used evaluating close structural analogues. This should be reflected in the paper.

2.           There are a couple of papers published on the reactivity prediction of different Michael acceptors including acrylamides and propyneamides (see e.g. Table 2 in ref. 35). The authors should discuss their results in the context of these data.

a.           One key question is that how the results of extended BIreactive do compare to earlier results published on propyneamides. This analysis might demonstrate the advantages of the methodology over other approaches.

b.           It would be interesting to discuss how the results of extended BIreactive do compare to that obtained for acrylamides.  This analysis might help the readers to understand the preference of one or the other warhead in a particular discovery program.

Author Response

.           The authors showed that electrophilicity indices are less useful for the prediction of reactivity for this class of compounds. This observation, however, is not new. There is a general consensus in the literature that electrophilicity index can be only used evaluating close structural analogues. This should be reflected in the paper.

Reply: Thank you for your comment, we referred to a literature example (10.1021/tx100172x) in the manuscript.

  1. There are a couple of papers published on the reactivity prediction of different Michael acceptors including acrylamides and propyneamides (see e.g. Table 2 in ref. 35). The authors should discuss their results in the context of these data.

Reply: We had a look at these manuscripts but there is none, where a systematic assessment of propyne amides was done. Most reactivity predictions are done for acrylamides. If there are propyne amides reported, it is mostly only the measured GSH value. We are to our knowledge the very first group that had such a thorough look at the prediction of the reactivity of propyne amides, so a comparison is hard to do.

  1. One key question is that how the results of extended BIreactive do compare to earlier results published on propyneamides. This analysis might demonstrate the advantages of the methodology over other approaches.

Reply: As mentioned before, we did not come across literature where the reactivity of (more than a few) propyne amides was reported.

  1. It would be interesting to discuss how the results of extended BIreactive do compare to that obtained for acrylamides.  This analysis might help the readers to understand the preference of one or the other warhead in a particular discovery program.

Reply: This is a very good suggestion. We discussed this internally and came to the conclusion, that this should be done in a separate paper. In this paper we want to focus on propyne amides and not water down the results we have achieved here.

Reviewer 3 Report

Review on MDPI Pharmaceuticals Manuscript (#2070207):

BIreactive: Expanding the Scope of Reactivity Predictions to 2 Propyneamides

The paper - written by M. R. Hermann and co-workers - describes computational predictive tools for assassment of propyneamides’ reactivity, which is a viable electrophilic warhead chemotype for targeted covalent inhibitors. The scientific topic of the manuscript might be of high interest by itself and furthermore, a truely reliable in silico prediction method would be heartfully welcomed in the field of covalent drug discovery, however, the work in the present form does not present enough novelty. Nonetheless, presentation of high amount of theoretical calculation and experimental results together would demonstrate a great scientific value, however, the prediction methodologies and correlation analysis needs of further improvents. Thus, I would recommend the paper for major revision, addressing the following concerns:

Majors issues:

1.        There are several previously published works dealing with in silico predictions, especially for Michael-acceptors. Schwöbel and co-workers provided a reliable predictive tool (Chem. Res. Toxicol. 2010, 23, 1576–1585) which can be applied even for propyne derivatives. What additinal value is presented here compared to the wide literature background? Furthermore, the method should be demostrated in a prospective manner, where real in silico prediction is provided and analyzed its performance by conventional statistical tools.

2.        If you constrain the critical C–S distance with 2.4A, how did you confirm, whether the structure you optimised is truely belongs tot he TS-geometry and not just a local maximum? In this part a seperated Figure or an extended „Figure 2” showing up all the possible (and investigated) directions of the nucleophilic attack would be benefitial. Furthermore the direction of the attack should be in line with the LUMO of the initial propyne compounds. Thus, showing the LUMO orbitals or the overlapping HOMO-LUMO orbitals in differents sets of possible TS-geometries would be also recommended.

3.        To calculate such thermodynamic parameters of a reaction the applied method and level of theory (ωB97XD/cc-pVDZ) could be a limiting factor due to its relatively inappropriate estimation of the enthalpic component. There are detailed investigations and comparisions of applicability of different DFT methods(https://doi.org/10.1063/1.4949536, https://doi.org/10.1039/C0CP02984J, https://doi.org/10.1021/ol701872z), suggesting there are far better choises for DFT calculations. Even if the large number of calculations needs of reducing the level of theory due to the enlarged time-consumption, one can confirm accuracy via combination of lower level geometry optimization and higher level single-point energy calculation (i.e.cc-pVDZ vs. cc-pVQZ), where lower level of theory ensure the estimation of entropic contribution and higher level of theory can refine the enthalpic component. For example the pair of 6-311G+(d,p) and 6-311++G(3df,3pd) is applied for similar type of calculations (https://doi.org/10.1007/s10822-020-00371-5) to demonstrate this methodology useful for in silico estimation of electrophilic reactivity and moreover the prediction of on-target efficacy of covalent probes. This refinement of the enthalpic component would be especially important because the calculations were even simpified (and possibly distorted) by trunctation algorithm. How the authors can rationalize the errors derived from the trunctation algorithm and the low level of theory? I would recommend to perform high-level calculations or even applying the abovementioned enthalpic correction for some representative compouns. I can accept if the applied level of calculations is suitable enough for reactivity prediction, however, it would be recommended to demonstrate that even higher level or more sophisticated refinement protocols are not providing significantly better estimation ot thermodynamical components.

4.        Investigation reaction route via Michael-type conjugate addition on Cβ is feasible, however propyne-derivatives are can either reactive for direct addition according to the Markovnyikov rule, which results in different product structure (involving rather Cα in reaction). Did the authors calculate direct addition TS either? If not, how can someone exclude that reaction meachanism?

5.        12 propyneamide fragments were synthesized out of the 28 designed structures. What about the additional 16? Were they problematic to synthesize, or what is the reason to shorten such a even originally small set of compounds? Even more what is the reason that „11 half-lives with GSH could be measured” of the 12 synthesized probes?

6.        Evaluation of GSH-assay should be described in more details, in the present format several methodoligical questions can be arised. It is not clear, that for what purpose the aqueous stability was assassed? Was it apply to corrigate the GSH-based kobs or used just as some additinal characterization parameter? Why the stability of the compounds is not reported at all, even though the measurment is described in details (even sampling time intervals are defined)? Did stability evaulation include linear regression based on 3 datapoints (according to the fact that 3 time was defined for sampling)?

7.        It is mentioned that „half-lives are calculated using Arrhenius plots”, however classical Arrhenius plots defined as connection between ln(k) and 1/T. It is not clear how Arrhenius plots were applied to calculate kinetic parameters. What type of regression algorithm was applied to calculate kinetic constant? What type of plots were analyzed and what kind of kinetic equations were utilized for experimental set up?

8.        Figure 3: Left-hand side (LHS) structures and even the whole library is well-composed and properly designed, would be really useful to investigate in silico methods’ predictive power. However, it is not clear what was the reason to present c, d, f and g LHS structures if they were not involved in the GSH-assay and later a much larger extension of the propyneamide compounds was carried out (Figure5d).

9.        The order of reactivity (H > Me ~ Ph > tBu) should be rationalized by easily available molecular descriptors based on the initial state geometries (i.e. Hammett-sigma, electron-density on theoretically attached H-atom as H–H / H–Me / H–Ph / H–tBu).

10.     After extension of the small benchmarking library, it is not clear why only „d” LHS-components were evaluated in GSH-assay. The conclusion would be more reliable if we could see a wider set of compounds. In this work 11×7 compounds (instead of the presented 11!) would be more comprehensive, applying all LHS (Figure 3) combined with all RHS (Figure 6).

Minor issues:

1.        Introduction about the importance of covalent inhibitors should include more up-to-date literature background including at least one fresh review or perspective about TCIs. (i.e. https://doi.org/10.1021/acs.jmedchem.1c02134)

2.        In the second paragraph the reversible TCI-development should be supported by literature reference, while the irreversible mechanism of action is supported by 20-21 references, however, I would recommend to include here all the latter cited references (22-24).

3.        In the same paragraph the series of approved TCIs might demand for better clafirication of the targeted kinases. More detailed and/or better organized text, or a more informative Figure 1 would be recommended (or both).

4.        In the next paragraph: „fine tune warhead reactivity by reliable predictions”. The following reference is strongly relevant here, thus, it is recommended to cite (in addition to the original 26-27 references): https://doi.org/10.1007/s10822-020-00371-5

5.        In the same paragraph (3rd): „often used as a measure of the reactivity for warheads within covalent binding compounds”. Insert here some relevant references, such as the 26-27 citations or some of the followings: https://doi.org/10.1016/j.ejmech.2018.10.010, https://doi.org/10.1016/j.drudis.2020.03.016, https://doi.org/10.1021/acs.jmedchem.0c00749, https://doi.org/10.1021/acs.jmedchem.5b01018, https://doi.org/10.1080/10659360500204152, https://doi.org/10.1021/tx100172x

6.        What restrictions or rules were applied for generating virtual library of propyneamides starting from the PubChem dataset?

7.        What type of „internal matched molecular pair algorithm” was applied to generate variations of noncovalent part? The authors should describe this method in more details, and even to refer to other MMP approach would be needed (https://doi.org/10.1016/j.csbj.2016.12.003).

8.        In the description of GSH-assay „mM” instead of „mmol/l” would be recommended. Even more mentioning the „kinetic HPLC/UV/MS based assay” would need for literature references where this methodology is explained and demonstrated in details: 26 ref (Flanagan et al.) and https://doi.org/10.1016/j.ejmech.2018.10.010

9.        Numbering of supplementary figures is not correct. The first mentioned is FigS5, then FigS1-S3 and much later the FigS4. Please fix the order and the numbering of the supplementary materials.

10.     In the main text: „the electrophilicity index was calculated for the moelcules…”. Please clarify which type of electrophilicity calculation was applied?

11.     Figure 4D/5D: The direct correlation between the activation Gibbs free energy and the reaction Gibbs free energy is well-known from previous works (https://doi.org/10.1038/nchem.291

12.     There are duplicated references, such as: 31-52 and 32-36.

Round 2

Reviewer 1 Report

The Authors have not completely answered to my suggestions

Reviewer 3 Report

I have reviewed the changes made by the authors and the responses that they made. I still consider the scientific topic of the manuscript as of high interest in the field of covalent drug discovery, and the author’s responses are very detailed and I think that the authors have justified the value of their work. The authors also made siginficant changes with the manuscript including important comments and additinal explanations within the text and even the supplementary information.Therefore, in my opinion the article could be accepted without further changes review.